# FedBCGD: Communication-Efficient Accelerated Block Coordinate Gradient Descent for Federated Learning

## ABSTRACT

Although federated learning has been widely studied in recent years, there are still high overhead expenses in each communication round for large-scale models such as Vision Transformer. To lower the communication complexity, we propose a novel Federated Block Coordinate Gradient Descent (FedBCGD) method for communication efficiency. The proposed method splits model parameters into several blocks including a shared block and enables uploading a specific parameter block by each client during training, which can significantly reduce communication overhead. Moreover, we also develop an accelerated FedBCGD algorithm (called FedBCGD+) with client drift control and stochastic variance reduction. To the best of our knowledge, this paper is the first parameter block communication work for training large-scale deep models. We also provide the convergence analysis for the proposed algorithms. Our theoretical results show that the communication complexities of our algorithms are a factor $1/N$ lower than those of existing methods, where $N$ is the number of parameter blocks, and they enjoy much faster convergence results than their counterparts. Empirical results indicate the superiority of the proposed algorithms compared to state-of-the-art algorithms.

## CCS CONCEPTS

• **Theory of computation** → *Distributed algorithms*.

## KEYWORDS

Federated Learning, Efficient Communication, Block Coordinate Gradient Descent

## 1 INTRODUCTION

Federated Learning (FL) is an emerging machine learning paradigm, which aims at achieving collaborative model training among multiple parties to preserve data privacy. Federated learning achieves model training by training models locally on client devices and then uploading them to a central server for model aggregation [28]. Compared to centralized learning in a data center [11], the parallel computing clients of federated learning have private data stored in them and communicate remotely with a central server. The clients are responsible for local training, while the central server in charge of aggregating the models uploaded by each client. Currently, federated learning has been widely applied in different fields such

*ACM MM, 2024, Melbourne, Australia*
© 2024 Copyright held by the owner/author(s). Publication rights licensed to ACM.
ACM ISBN 978-x-xxxx-xxxx-x/YY/MM
https://doi.org/10.1145/nnnnnnn.nnnnnnn

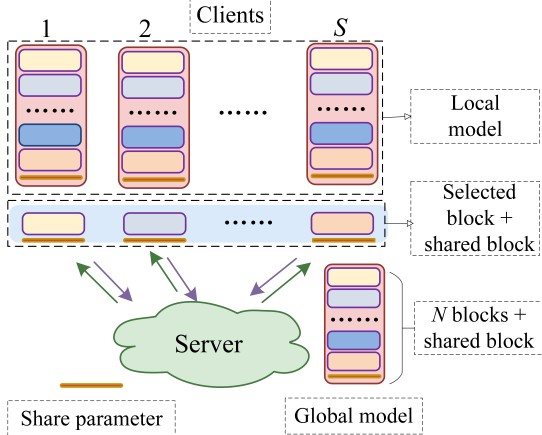

**Figure 1: The diagram of the proposed FedBCGD framework, where $S \geq N$, $S$ and $N$ are the numbers of clients and parameter blocks, respectively.**

as mobile intelligence devices, medical health, and financial risk control [3, 5, 35].

In mainstream federated learning frameworks, the communication between clients and their server is slow, costly, and unreliable [19]. In recent years, large models such as BERT and ChatGPT [4, 7] have emerged, leading to an exponential increase in the model size and data volume on FL clients. The upload of these large models further escalates the cost of communication in FL. To significantly lower the communication complexity, this paper proposes a novel method for federated learning, called Federated Block Coordinate Gradient Descent (FedBCGD), which is based on block coordinate descent (BCD) methods [38].

In FL, the upload speed of the client model is more than a hundred times slower than the download speed, so this paper mainly resolves the issue of upload communication cost. As shown in Figure 1, we divide the model parameter $x$ into $N$ blocks and $x_s$, i.e., $x = [x_{(1)}^\top, \ldots, x_{(N)}^\top, x_s^\top]^\top$, where $x_s$ denotes the shared parameters in each client (usually the parameters of the last layer classifier, and their number is small but important, [26] suggests that the deeper the model, the greater the variance of the parameters. In federated learning, it is often the parameters in the last layer of the classifier that are most important and have a very small number of covariates (0.01% of the overall number in ResNet-18)). Each client is responsible for optimizing one selected block $x_{(j)}$ and $x_s$. After local training for all model parameters, the updated parameter block $x_{(j)}$ and $x_s$ are sent to the central server, which takes the average aggregation of parameters for different parameter blocks to get the complete model.

The initial idea is to require each client to perform local updates only on the specified parameter block $x_{(j)}$ and $x_s$ while freezing

**Table 1: Comparison of the communication complexities and communication overheads of different algorithms in the $\mu$-strongly convex setting, where $\sigma$ is the variance of stochastic gradients, $G$ is heterogeneity due to client data distribution, $S$ is the number of participating clients, $K=S/N$, $K$ is the number of clients involved in each parameter block, $N$ is the number of parameter blocks, and $T$ is the number of local training iterations. The number of floats sent per round by FedAvg is $d$, and $O$ describes the worst-case complexity of different algorithms, where $\alpha = \frac{1}{1-\lambda}$. In non-convex settings, $\tau$ is the second-order heterogeneity (see [16]), $G$ is the first-order heterogeneity, $F \coloneqq f(x^0) - f^\star$, and $f^\star$ is a minimum value of Problem (1) below.**

| Algorithm | Strongly convex Communication complexity | Non-convex Communication complexity | Client sample | Stochastic Gradient | Floats sent per round |
|---|---|---|---|---|---|
| FedAvg [28] | $O\left(\frac{\sigma^2+G^2}{\mu ST\epsilon} + \frac{\sigma+G}{\mu\sqrt{\epsilon}} + \frac{\beta}{\mu}\log\frac{1}{\epsilon}\right)$ | $O\left(\frac{\beta\sigma^2}{TS\epsilon^2} + \frac{\sqrt{\beta}G+\sqrt{\frac{\beta}{T}}\sigma}{\epsilon^{\frac{3}{2}}} + \frac{F\beta}{\epsilon}\right)$ | Yes | Yes | $d$ |
| **FedBCGD (ours)** | $O\left(\frac{\sigma^2+G^2}{\mu ST\epsilon} + \frac{\sigma+G}{\alpha\mu N\sqrt{\epsilon}} + \frac{\beta}{\mu N}\log\frac{1}{\epsilon}\right)$ | $O\left(\frac{\beta\sigma^2}{TS\epsilon^2} + \frac{\sqrt{\beta}G+\sqrt{\frac{\beta}{T}}\sigma}{N\epsilon^{\frac{3}{2}}} + \frac{F\beta}{N\epsilon}\right)$ | Yes | Yes | $d/N$ |
| SCAFFOLD [17] | $O\left(\frac{\sigma^2}{\mu ST\epsilon} + \frac{\sigma}{\mu\sqrt{\epsilon}} + \left(\frac{M}{S}+\frac{\beta}{\mu}\right)\log\frac{1}{\epsilon}\right)$ | $O\left(\frac{\beta\sigma^2}{TS\epsilon^2} + \frac{\sqrt{\frac{\beta}{T}}\sigma}{\epsilon^{\frac{3}{2}}} + \frac{\beta F}{\epsilon}\left(\frac{M}{S}\right)^{\frac{2}{3}}\right)$ | Yes | Yes | $2d$ |
| FedLin [29] | $O\left(\frac{\beta}{\mu}\log\frac{1}{\epsilon}\right)$ | —— | No | No | $2d$ |
| S-Local-GD [10] | $O\left(\frac{\beta}{\mu}\log\frac{1}{\epsilon}\right)$ | —— | No | No | $2d$ |
| CE-LSGD [31] | —— | $O\left(\frac{GF\tau}{M\epsilon^{\frac{3}{2}}}\right)$ | Yes | Yes | $3d$ |
| BVR-L-SGD [30] | —— | $O\left(\frac{F\tau}{\epsilon} + \frac{F\beta}{\sqrt{T}\epsilon} + \frac{\sigma^2}{MT\epsilon} + \left(\frac{\sigma F\beta}{MT\epsilon}\right)^{\frac{3}{2}}\right)$ | Yes | Yes | $3d$ |
| **FedBCGD+ (ours)** | $O\left(\left(\frac{M}{S} + \sqrt{\frac{\beta}{\mu}}\right)\log\frac{1}{\epsilon}\right)$ | $O\left(\frac{\beta F}{\epsilon}\left(\frac{M}{S}\right)^{\frac{2}{3}}\frac{1}{N}^{\frac{1}{3}}\right)$ | Yes | Yes | $2d/N$ |

the remaining parameter blocks (called FedBCGD_freezing). After local training, the specified parameter blocks would be uploaded for model aggregation. However, due to a large drift between parameter blocks, such scheme often results in bad convergence in our experiments (see Figure 5 for details). More specifically, only updating certain parameter blocks locally results in a large gap between the updated parameter blocks and other freezing parameter blocks, and it is not possible to establish good connections between parameter blocks during the server-side aggregation process.

Therefore, we propose a novel FedBCGD method to address these issues. In the proposed algorithm, we employ stochastic gradient descent to update all parameters instead of parameter freezing during local training, but only transmit two specified parameter blocks ($x_{(j)}$ and $x_s$) during the upload process. In addition, to compensate for some missing parameters in block parameter transmission, we add parameter block momentum on the server side. This algorithm design maintains the advantages of low communication costs and has demonstrated a significantly improved convergence speed in our experiments (see Figure 5 for details). Moreover, adding one shared parameter block in each client can significantly improve accuracy performance. However, due to the impact of data heterogeneity, it still leads to inconsistent update directions between parameter blocks, called parameter block drift, resulting in poor performance of the aggregated model. Thus, we also propose an accelerated version (called FedBCGD+) to address data heterogeneity. The main difference between FedBCGD and BCD is that FedBCGD incorporates shared one small parameter block and updates all model parameters in each client (i.e., no parameter freezing), while BCD only updates one parameter block in each iteration.

**Our motivations and contributions:** To address these issues such as communication effectiveness, acceleration, theoretical guarantees and parameter block drift, we design a novel federated block coordinate descent framework FedBCGD and its acceleration variant FedBCGD+ for training large-scale deep models such as Vision Transformer. The main contributions of this work are listed as follows:

• **Novel Federated Learning Paradigm:** We propose the first block coordinate descent algorithm FedBCGD for horizontal FL. FedBCGD demonstrates remarkable communication efficiency in distributed learning scenarios. That is, this paper presents the first block coordinate descent algorithm for horizontal federated learning. Moreover, we also introduce an accelerated version, FedBCGD+, which exhibits an even faster convergence rate while maintaining high communication efficiency.

• **Convergence Analysis:** We provide a thorough analysis of the convergence properties of the proposed FedBCGD algorithm and its accelerated version, FedBCGD+. By investigating the impact of partitioned parameter blocks, the number of clients, and the local training rounds, we provide valuable insights into their convergence behavior. From a practical perspective, FedBCGD+ achieves faster convergence than FedBCGD, and it is proved faster from a theoretical perspective. Moreover, FedBCGD+ has a much lower communication complexity than existing algorithms in strong convexity settings (e.g., $O\left(\left(\frac{M}{S} + \sqrt{\frac{\beta}{\mu}}\right)\log\frac{1}{\epsilon}\right)$ for FedBCGD+ vs. $O\left(\frac{\sigma^2}{\mu ST\epsilon} + \frac{\sigma}{\mu\sqrt{\epsilon}} + \left(\frac{M}{S}+\frac{\beta}{\mu}\right)\log\frac{1}{\epsilon}\right)$ for SCAFFOLD [17]. Furthermore, we can achieve a significant lower communication complexity of $O\left(\frac{\beta F}{\epsilon}\left(\frac{M}{S}\right)^{2/3}\frac{1}{N}^{1/3}\right)$ in the non-convex setting, compared to that

 

of SCAFFOLD, $O\left(\frac{\beta\sigma^2}{TS\epsilon^2} + \frac{\sqrt{\frac{\beta}{T}}\sigma}{\epsilon^{3/2}} + \frac{\beta F}{\epsilon}\left(\frac{M}{S}\right)^{2/3}\right)$. In other words, the communication complexities of our algorithms are a factor $1/N$ lower than those of existing methods, where $N$ is the number of parameter blocks.

• **Overcoming Data Heterogeneity :** The convergence of federated learning algorithms is hindered by two sources of high variance: (i) heterogeneous clients, and (ii) the noise from local stochastic gradients. We propose two sets of control variance variables to reduce client heterogeneity and the noise variance of the local gradients in FedBCGD+. And we demonstrate the validity of the two sets of control variables through theory and experiment.

## 2 RELATED WORK

In this section, we mainly review existing federated learning and block coordinate descent methods.

• **Local Training:** Local Training (LT) is a communication-acceleration technique for FL [28]. One key challenge in LT is client drift, where the local model of each client gradually approaches the minimum of its own local cost function $f_i$ after multiple local GD steps. To address this issue, SCAFFOLD [17] is proposed, which was to incorporate control variates to correct for client drift and ensure linear convergence to the exact solution. Subsequent algorithms such as S-Local-GD [10] and FedLin [29] also aimed to provide similar convergence properties.

The analysis of algorithms for non-convex federated learning can be classified into several approaches. SCAFFOLD [17] is the first federated algorithm capable of eliminating client data heterogeneity. However, its convergence speed is still affected by stochastic gradients, achieving only a convergence rate of $O(1/\epsilon^2)$. MIME [16] is essentially a combination of local SGD and variance reduction techniques as in SVRG [15], with a derived communication complexity of $O(1/\epsilon^{3/2})$. BVR-L-SGD [30] assumed second-order data heterogeneity and achieved a communication complexity of $O(1/\epsilon)$ with full client participation. The two-sided momentum (STEM) algorithm [18] can also attain a communication complexity of $O(1/\epsilon)$ with full client participation. Inspired by the Storm algorithm [6], CE-LSGD [31] can achieve a communication complexity of $O(1/\epsilon^{3/2})$ with partial client participation and $O(1/\epsilon)$ when all clients participate.

• **Block Coordinate Descent Methods:** The block coordinate descent method is one of the most successful algorithms in the field of big data optimization. BCD is based on the strategy of updating a single coordinate or a single block of coordinate of a vector of variables at each iteration, which usually significantly reduces the memory requirements as well as the arithmetic complexity of a single iteration. The effectiveness of the BCD method for training deep neural networks (DNNs) has been demonstrated in recent years [40]. However, due to the highly non-convex nature of deep neural networks, its convergence is difficult to maintain. In addition, BCD can be easily implemented in a distributed and parallel manner [27, 34]. Liu et al. [25] proposed a vertical federated learning [24] framework (FedBCD) for distributed features, in which parties share only the internal product of model parameters and raw data for each sample during each communication. Unlike the above works, this paper proposes the first block coordinate descent algorithm for horizontal federated learning. Horizontal federated learning

is applied to scenarios where the client's datasets have the same feature space and different sample spaces [39].

• **Communication-efficient Federated Learning:** Communication efficient Federated Learning algorithms can be divided into two categories, quantization and sparsification compression methods. The classical Federated Learning quantization method is proposed by Reisizadeh [33], which is a cycle averaging and quantization processing method named FedPAQ, and the quantization compression generally belongs to the unbiased compressions. While sparsification methods include *top-k* and *rand-k* methods [36], *top-k* method is a biased compression method that uploads the gradient at the first $k$ large positions in the gradient to the server, while *rand-k* method is an unbiased compression method that uploads the gradient at random $k$ positions to the server. FedBCGD is different from all of the above methods and utilizes the idea of block gradient descent to address federated efficient communication, in addition to the above mentioned compression method that allows for secondary compression of our transferred block gradient to achieve more efficient communication, which is demonstrated in the following experiment.

## 3 COMMUNICATION-EFFICIENT BLOCK COORDINATE GRADIENT DESCENT FEDERATED LEARNING

In this section, we propose a new communication-efficient block coordinate gradient descent federated learning algorithm FedBCGD, and its pseudocode is given in Algorithm 1. We formalize the federated learning problem as the minimization of a sum of stochastic functions:

$$\min_{\boldsymbol{x}\in\mathbb{R}^d}\left\{f(\boldsymbol{x}) := \frac{1}{M}\sum_{i=1}^{M}\left(f_i(\boldsymbol{x}) := \frac{1}{n_i}\sum_{v=1}^{n_i}f_i\left(\boldsymbol{x};\zeta_{i,v}\right)\right)\right\}, \quad (1)$$

where the function $f_i$ denotes the loss function on client $i$, $M$ is the number of clients, $n_i$ is the number of data points in client $i$, and $\{\zeta_{i,1}, \ldots, \zeta_{i,n_i}\}$ denote the local data of the $i$-th client. In this paper, we assume that $f_i$ is a $\beta$-smooth function.

### 3.1 The proposed FedBCGD Algorithm

We firstly divide the global model $\boldsymbol{x}$ into $N$ blocks of parameters and one shared block, each of which can have a different number of parameters,
$$\boldsymbol{x} = \left[\boldsymbol{x}_{(1)}^{\top}, \ldots, \boldsymbol{x}_{(N)}^{\top}, \boldsymbol{x}_s^{\top}\right]^{\top}. \quad (2)$$
We divide the sampled $S = N \cdot K$ clients into $N$ client blocks with $K$ clients in each client block (see Figure 2). These $N$ parameter blocks are distributed to the selected $N$ client blocks, where each parameter block will be optimized by $K$ clients. Due to significant differences in communication capabilities among different clients, parameter blocks with smaller parameter values can be assigned to clients with poorer communication capabilities, while parameter blocks with larger parameter values can be assigned to clients with better communication capabilities. This prevents clients with the smallest resources from becoming bottlenecks in federated learning. We define $\boldsymbol{x}_{k,j}$ as the local parameters of $k$-th client in $j$-th client block (as client$_{k,j}$). Each client performs $T$ local stochastic gradient steps on its respective client block, by using a minibatch in each iteration as follows:

$$\boldsymbol{x}_{k,j}^{r,t+1} = \boldsymbol{x}_{k,j}^{r,t} - \eta \nabla f_{k,j}(\boldsymbol{x}_{k,j}^{r,t};\zeta), \quad (3)$$

**Algorithm 1** FedBCGD

---

1: **Initialize** $x_i^{0,0} = x^{init}, \forall i \in [M]$.

2: **Divide** the model parameters $x$ into $N+1$ blocks.

3: **for** $r = 0, ..., R$ **do**

4:     **Client:**

5:     **Sample** clients $\mathcal{S} \subseteq \{1, \dots, M\}, |\mathcal{S}| = N \cdot K$;

6:     **Divide** the sampled clients into $N$ client blocks;

7:     **Communicate** $(x^r)$ to all clients $i \in \mathcal{S}$;

8:     **for** $j = 1, \dots, N$ client blocks in parallel **do**

9:         **for** $k = 1, \dots, K$ clients in parallel **do**

10:             **for** $t = 1, \dots, T$ local update **do**

11:                 Compute batch gradient $\nabla f_{k,j}(x_{k,j}^{r,t}; \zeta)$,

12:                 $x_{k,j}^{r,t+1} = x_{k,j}^{r,t} - \eta \nabla f_{k,j}(x_{k,j}^{r,t}; \zeta)$;

13:             **end for**

14:             Send $x_{k,j,(j)}^{r,T}, x_{k,j,s}^{r,T}$ to server;

15:         **end for**

16:     **end for**

17:     **Server:**

18:     **for** $j = 1, \dots, N$ Blocks in parallel **do**

19:         Block $j$ computes,

20:         $x_{(j)}^r = \frac{1}{K}\sum_{k=1}^{K} x_{k,j,(j)}^{r,T}; v_{(j)}^r = \lambda v_{(j)}^{r-1} + x_{(j)}^r - x_{(j)}^{r-1}$;

21:         $x_{(j)}^r = x_{(j)}^r + v_{(j)}^r$,

22:     **end for**

23:     $x_s^r = \frac{1}{NK}\sum_{j=1}^{N}\sum_{k=1}^{K} x_{k,j,s}^{r,T}; v_s^r = \lambda v_s^r + x_s^{r-1} - x_s^{r-1}$;

24:     $x_s^r = x_s^r + v_s^r; x^r = \left[x_{(1)}^{r\top}, \dots, x_{(N)}^{r\top}, x_s^{r\top}\right]^\top$;

25:     $v^r = \left[v_{(1)}^{r\top}, \dots, v_{(N)}^{r\top}, v_s^{r\top}\right]^\top$;

26: **end for**

---

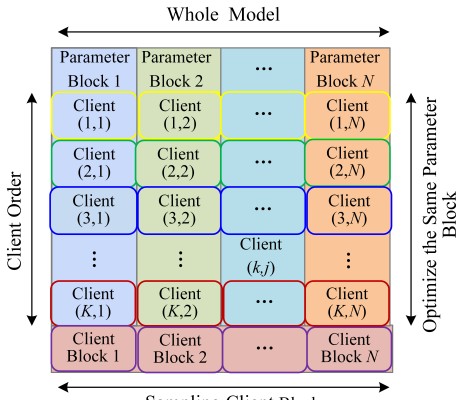

**Figure 2: The client parameter block allocation in FedBCGD. For the sake of convenience, we suppose $S = N \cdot K$ clients are sampled and divided into $N$ client blocks, i.e., $K$ clients for each client block. The clients in the $i$-th client block are responsible for optimizing the upload parameter block $i$.**

where $x_{k,j}^{t+1}$ is the $t+1$-th local update whole parameter of client$_{k,j}$, and $x_{k,j,(j)}^{t+1}$ is the $j$-th parameter block of client$_{k,j}$. $\nabla f_{k,j,(j)}$ is the $j$-th gradient block in client$_{k,j}$ (see Figure 2). The local client of

FedBCGD is used to update all model parameters $x$ and send the selected parameter block $x_{(j)}$ and $x_s$ to the server.

Below, we will describe the proposed server-side aggregation operation. For the $k$-th client of the $j$-th parameter block, it sends the parameter block $x_{k,j,(j)}^{r,T}$ and $x_{k,j,s}^{r,T}$ after $T$ times local update to server. The central server performs separate aggregation operations on $x_{(j)}$ and $v_{(j)}$ for the $j$-th parameter block in Lines 20-22 of Algorithm 1. Next, we update the shared parameter block in Lines 24-26 of Algorithm 1. Finally, all the parameter blocks are combined into a complete model, $x^r = \left[x_{(1)}^{r\top}, \dots, x_{(N)}^{r\top}, x_s^{r\top}\right]^\top$, and the momentum term is $v^r = \left[v_{(1)}^{r\top}, \dots, v_{(N)}^{r\top}, v_s^{r\top}\right]^\top$. Before the next iteration starts, the client transfers all model parameters $x^r$ to the selected client and tells the client which model parameter block needs to be uploaded.

$v_{(j)}^r$ is the $j$-th block of the momentum term $v^r$, and $\lambda$ is the momentum parameter. The momentum term $v_{(j)}^r$ considers the model's continuous updates over time, making the updating process smoother. More specifically, it remembers and utilizes the direction and speed of previous model parameter updates, thereby accelerating the convergence speed of the model.

## 3.2 Our FedBCGD+ Algorithm

The FedBCGD+ algorithm is an extension of our FedBCGD algorithm based on the principles of variance reduction in SVRG [15]. And its details are presented in the Appendix. Note that the server-side updates in FedBCGD+ are consistent with FedBCGD, while the new proposed client-side update of our FedBCGD+ algorithm is formulated as follows:

$$x_{k,j}^{r,t+1} = \underbrace{x_{k,j}^{r,t} - \eta \nabla f_{k,j}(x_{k,j}^{r,t}; \zeta)}_{\text{Stochastic Gradient Descent}} + \underbrace{\eta c - \eta c_{k,j}}_{\text{Client Drift Control Variate}}$$
$$+ \underbrace{\eta \nabla f_{k,j}(x^r) - \eta \nabla f_{k,j}(x^r; \zeta)}_{\text{Stochastic Variance Reduction}}. \quad (4)$$

That is, each client-side update consists of a stochastic gradient descent (SGD) term, one client drift control variate term and a variance reduction term, which is different from all existing works such as [17].

FedBCGD+ maintains a state for each client (the client control variate $c_i$) and the server (the server control variate $c$). Here, $c_{k,j}^+ = \nabla f_{k,j}(x^r)$, and we need to send $x_{k,j,(j)}^{r,T}, \Delta c_{k,j,(j)} = c_{k,j,(j)}^+ - c_{k,j,(j)}$, $\Delta c_{k,j,s} = c_{k,j,s}^+ - c_{k,j,s}$ to the server, $c_i = c_i^+$. We update $c$ on the server-side as follows:

$$c_{(j)} = c_{(j)} + \frac{1}{M}\sum_{k=1}^{K} \Delta c_{k,j,(j)}, \quad (5)$$

$$c_s = c_s + \frac{1}{MN}\sum_{j=1}^{N}\sum_{k=1}^{K} \Delta c_{k,j,s}, \quad (6)$$

$$c = \left[c_{(1)}^\top, \dots, c_{(N)}^\top, c_s^\top\right]^\top. \quad (7)$$

The key of our FedBCGD+ algorithm for improving the convergence speed is based on the following observation. The convergence of federated learning algorithms is hindered by two sources of high variance: (i) the global server aggregation step and multiple local

updates, which are exacerbated by client heterogeneity, and (ii) the noise from local client-level stochastic gradients.

In the local update in Eq. (4), the first term involves stochastic gradient descent, the second term incorporates client heterogeneity control inspired by SCAFFOLD [17], and the third term adopts one stochastic variance reduction technique as in SVRG [15] to reduce the variance of stochastic gradients. By integrating these three components, our algorithm effectively addresses the challenges posed by heterogeneous clients and noisy local gradients, leading to a significant improvement in the convergence speed during the federated learning process. Compared with existing algorithms such as SCAFFOLD, and our FedBCGD, FedBCGD+ has a faster convergence rate, as shown in the following theoretical results.

## 4 THEORETICAL GUARANTEES

In this section, we provide rigorous theoretical analysis for all the proposed algorithms, and the detailed proofs are included in the Appendix. The theoretical analysis of our FedBCGD algorithm is not a simple parallelization extension of the traditional BCD algorithm but an innovative theoretical analysis framework. Compared with related work, the two proposed algorithms have some theoretical advantages, including faster convergence rates and lower communication complexities. For the convenience of theoretical analysis, we ignore the shared block in the algorithms.

### 4.1 Theoretical Results of FedBCGD

THEOREM 1 (FEDBCGD). *For $\beta$-smooth functions $\{f_i\}$, which satisfy Assumptions 1-5 (see the Appendix for details), the output of FedBCGD has expected error smaller than $\epsilon$ for some values of $\eta, R$, where $R$ denotes the number of communication rounds, Com is the communication complexity (i.e., the product of the number of communication rounds and the floats sent per round) satisfying:*
**Strongly convex**: $\tilde{\eta} = \frac{\alpha\eta T}{4}, \tilde{\eta} \leq \frac{1}{8\beta}$, and

$$R = O\big(\frac{\sigma^2 + G^2}{\mu KT\epsilon} + \frac{\sigma + G}{\alpha\mu\sqrt{\epsilon}} + \frac{\beta}{\mu}\log\frac{1}{\epsilon}\big),$$

$$Com = O\big(\frac{\sigma^2 + G^2}{\mu ST\epsilon}d + \frac{\sigma + G}{\alpha\mu N\sqrt{\epsilon}}d + \frac{\beta}{\mu N}\log\frac{1}{\epsilon}d\big),$$

**Non-convex**: $\tilde{\eta} = \frac{1}{4}\alpha\eta T, \tilde{\eta} \leq \frac{1}{16\beta}, F := f\left(x^0\right) - f^{\star}$,

$$R = O\big(\frac{\beta\sigma^2}{TK\epsilon^2} + \frac{\sqrt{\beta}G + \sqrt{\frac{\beta}{T}}\sigma}{\epsilon^{\frac{3}{2}}} + \frac{F\beta}{\epsilon}\big),$$

$$Com = O\big(\frac{\beta\sigma^2}{TS\epsilon^2}d + \frac{\sqrt{\beta}G + \sqrt{\frac{\beta}{T}}\sigma}{N\epsilon^{\frac{3}{2}}}d + \frac{F\beta}{N\epsilon}d\big).$$

From Table 1, comparing the second term of communication complexity of FedAvg (i.e., $O\big(\frac{\sigma+G}{\mu\sqrt{\epsilon}}d\big)$), the term of FedBCGD is $O\big(\frac{\sigma+G}{\alpha\mu N\sqrt{\epsilon}}d\big)$, which is $N$ times significantly lower. As the number of blocks $N$ increases, FedBCGD can achieve a significantly lower communication complexity, and we will verify this in the experimental section (see Figure 4). The momentum parameter $\alpha$ here is equivalent to the server step size, and a larger server step size can accelerate convergence, as pointed out in [17].

### 4.2 Theoretical Results of FedBCGD+

THEOREM 2 (FEDBCGD+). *For $\beta$-smooth functions $\{f_i\}$, which satisfy Assumptions 1-5, the output of FedBCGD+ has expected error smaller than $\epsilon$ for some values of $\eta, R$, where $R$ and $Com$ satisfy:*
**Strongly convex**: $\tilde{\eta} = \frac{\alpha\eta T}{4}, \tilde{\eta} \leq \frac{1}{8\beta}$, and

$$R = O\big((\frac{M}{K} + \frac{\beta}{\mu})\log\frac{1}{\epsilon}\big), Com = O\big((\frac{M}{S} + \frac{\beta}{\mu N})d\log\frac{1}{\epsilon}\big),$$

**Non-convex**: $\tilde{\eta} = \frac{1}{4}\alpha\eta T, \tilde{\eta} \leq \frac{1}{16\beta}, F := f\left(x^0\right) - f^{\star}$,

$$R = O\big(\frac{\beta F}{\epsilon}\big(\frac{M}{K}\big)^{\frac{2}{3}}\big), \quad Com = O\big(\frac{\beta F}{\epsilon}\big(\frac{M}{S}\big)^{\frac{2}{3}}\frac{1}{N}d\big).$$

The communication complexity of FedBCGD is $O\big(\frac{\sigma^2+G^2}{\mu ST\epsilon}d + \frac{\sigma+G}{\alpha\mu N\sqrt{\epsilon}}d + \frac{\beta}{\mu N}\log\frac{1}{\epsilon}d\big)$ in the strongly convex setting. The main influence on the communication complexity is determined by the two parameters, $G$ (client heterogeneity) and $\sigma$ (noise of stochastic gradients). FedBCGD+ resolves these issues, and can achieve the communication complexity of $O\big((\frac{M}{S} + \frac{\beta}{\mu N})d\log\frac{1}{\epsilon}\big)$. When $N = \sqrt{\beta/\mu}$, and its communication complexity is $O\left(\left(\frac{M}{S} + \sqrt{\frac{\beta}{\mu}}\right)d\log\frac{1}{\epsilon}\right)$, which significantly improves the best-known result (see Table 1 for details). When $\sigma = 0$, the communication complexity of FedBCGD is also better than that of SCAFFOLD, $O\big((\frac{M}{S} + \frac{\beta}{\mu})d\log\frac{1}{\epsilon}\big)$. Without client sampling ($S = M$), the communication complexity of Fed-BCGD+ is $O\big(\sqrt{\frac{\beta}{\mu}}d\log\frac{1}{\epsilon}\big)$, which is much better than that of FedLin [29], $O\big(\frac{\beta}{\mu}d\log\frac{1}{\epsilon}\big)$. In the non-convex setting, the communication complexity of FedBCGD+ is $O\big(\frac{\beta F}{\epsilon}\big(\frac{M}{S}\big)^{2/3}N^{-1/3}d\big)$, which also is the best-known result (see Table 1). Without client sampling, the communication complexity of FedBCGD+ is $O\big(\frac{\beta F}{\epsilon}N^{-1/3}d\big)$, which is much better than that of CE-LSGD [31], $O\big(\frac{\beta F}{\epsilon}d\big)$. As the number of blocks $N$ increases, FedBCGD+ can also achieve a significantly lower communication complexity.

## 5 EXPERIMENTS

In this section, we conduct various experiments for convex and non-convex problems, and more results are reported in the Appendix.

### 5.1 Experimental Settings and Baselines

**Datasets:** We evaluate our algorithms on the CIFAR10 [20], CIFAR100 [20], Tiny ImageNet [21] and EMNIST datasets. We set up a total of 100 clients in the FL experiment with a participation rate of 10%. For the non-IID data setup, we model data heterogeneity by sampling label ratios $\rho$ from a Dirichlet distribution.
**Models:** To test the robustness of our algorithms, we use standard classifiers (including LeNet-5 [22], VGG-11, VGG-19 [37], and ResNet-18 [13]), Vision Transformer (ViT-Base) [8]. We divided the parameters of the model into 5 blocks or more blocks and provide the detailed parameter block division of the model in the Appendix.
**Methods:** We compare FedBCGD and FedBCGD+ with many SOTA FL baselines, including FedAvg [28], SCAFFOLD [17], FedAvgM [14], FedDC [9], FedAdam [32], and TOP-k [1], FedPAQ [33].
**Hyper-parameter Settings:** The initial learning rate is searched in {0.01, 0.03, 0.05, 0.1, 0.2, 0.3}, with a decay of 0.998 and a weight decay of 0.001 for each round.

**Table 2: Comparison of the average testing accuracy (%) over the last 10% rounds of each algorithm on CIFAR100, where the heterogeneity parameter is $\rho = 0.6$, total communication floats are $1000d$, and the number of blacks is $N = 5$. The number in brackets indicates the number of communication floats to reach the target accuracy. Note that centralised SGD refers to using SGD to train models on a single machine.**

| CIFAR100 | LeNet-5 (40%) | VGG-11 (48%) | ResNet-18 (54%) | VGG-19 (45%) |
|---|---|---|---|---|
| Centralised SGD | $53.7 \pm 0.2$ | $56.3 \pm 0.3$ | $62.2 \pm 0.1$ | $58.9 \pm 0.1$ |
| FedAvg [28] | $41.2 \pm 0.2$ ($558d$) | $48.7 \pm 0.4$ ($720d$) | $54.2 \pm 0.2$ ($927d$) | $47.6 \pm 0.1$ ($735d$) |
| FedAvgM [14] | $48.2 \pm 0.5$ ($277d$) | $51.7 \pm 0.6$ ($299d$) | $61.8 \pm 0.8$ ($398d$) | $56.0 \pm 0.3$ ($403d$) |
| FedAdam [32] | $46.2 \pm 0.8$ ($391d$) | $50.9 \pm 0.5$ ($597d$) | $53.9 \pm 0.4$ ($\infty$) | $58.7 \pm 0.2$ ($367d$) |
| SCAFFOLD [17] | $50.3 \pm 0.2$ ($214d$) | $47.9 \pm 0.2$ ($\infty$) | $52.3 \pm 0.2$ ($\infty$) | $58.3 \pm 0.5$ ($556d$) |
| FedDC [9] | $53.2 \pm 0.3$ ($302d$) | $48.2 \pm 0.2$ ($956d$) | $46.6 \pm 0.1$ ($\infty$) | $56.8 \pm 0.4$ ($321d$) |
| **FedBCGD (ours)** | $\mathbf{55.7 \pm 0.4}$ ($77d$) | $\mathbf{62.2 \pm 0.4}$ ($107d$) | $\mathbf{68.1 \pm 0.5}$ ($277d$) | $61.1 \pm 0.3$ ($206d$) |
| **FedBCGD+ (ours)** | $55.6 \pm 0.3$ ($\mathbf{75d}$) | $58.7 \pm 0.3$ ($\mathbf{105d}$) | $65.1 \pm 1.8$ ($\mathbf{154d}$) | $\mathbf{63.6 \pm 0.4}$ ($\mathbf{176d}$) |

**Table 3: Comparison of the average testing accuracy (%) over the last 10% rounds of each algorithm on CIFAR10, where the heterogeneity parameter is $\rho = 0.6$, total communication floats are $1000d$, the number of blacks is set to $N = 5$.**

| CIFAR10 | LeNet-5 (78%) | VGG-11 (83%) | ResNet-18 (88%) | VGG-19 (84%) |
|---|---|---|---|---|
| Centralised SGD | $83.1 \pm 0.2$ | $87.4 \pm 0.3$ | $90.1 \pm 0.1$ | $88.6 \pm 0.1$ |
| FedAvg [28] | $79.6 \pm 0.3$ ($498d$) | $83.3 \pm 0.7$ ($630d$) | $89.0 \pm 0.5$ ($698d$) | $84.9 \pm 0.7$ ($499d$) |
| FedAvgM [14] | $81.1 \pm 0.6$ ($360d$) | $83.7 \pm 0.4$ ($830d$) | $89.1 \pm 0.7$ ($882d$) | $87.4 \pm 0.5$ ($252d$) |
| FedAdam [32] | $78.3 \pm 1.2$ ($860d$) | $85.4 \pm 1.1$ ($478d$) | $81.1 \pm 1.3$ ($\infty$) | $87.5 \pm 0.9$ ($298d$) |
| SCAFFOLD [17] | $82.8 \pm 0.7$ ($540d$) | $86.9 \pm 0.6$ ($278$) | $89.0 \pm 0.4$ ($747d$) | $85.5 \pm 0.5$ ($358d$) |
| FedDC [9] | $83.0 \pm 0.2$ ($280d$) | $83.1 \pm 0.6$ ($866$) | $88.0 \pm 0.6$ ($1985d$) | $78.0 \pm 0.9$ ($\infty$) |
| **FedBCGD (ours)** | $\mathbf{84.7 \pm 0.7}$ ($249d$) | $\mathbf{88.4 \pm 0.7}$ ($292d$) | $\mathbf{92.1 \pm 0.3}$ ($398d$) | $\mathbf{87.8 \pm 0.4}$ ($\mathbf{117d}$) |
| **FedBCGD+ (ours)** | $83.5 \pm 0.3$ ($\mathbf{182d}$) | $88.3 \pm 0.4$ ($\mathbf{209d}$) | $90.3 \pm 0.5$ ($\mathbf{266d}$) | $87.1 \pm 0.4$ ($207d$) |

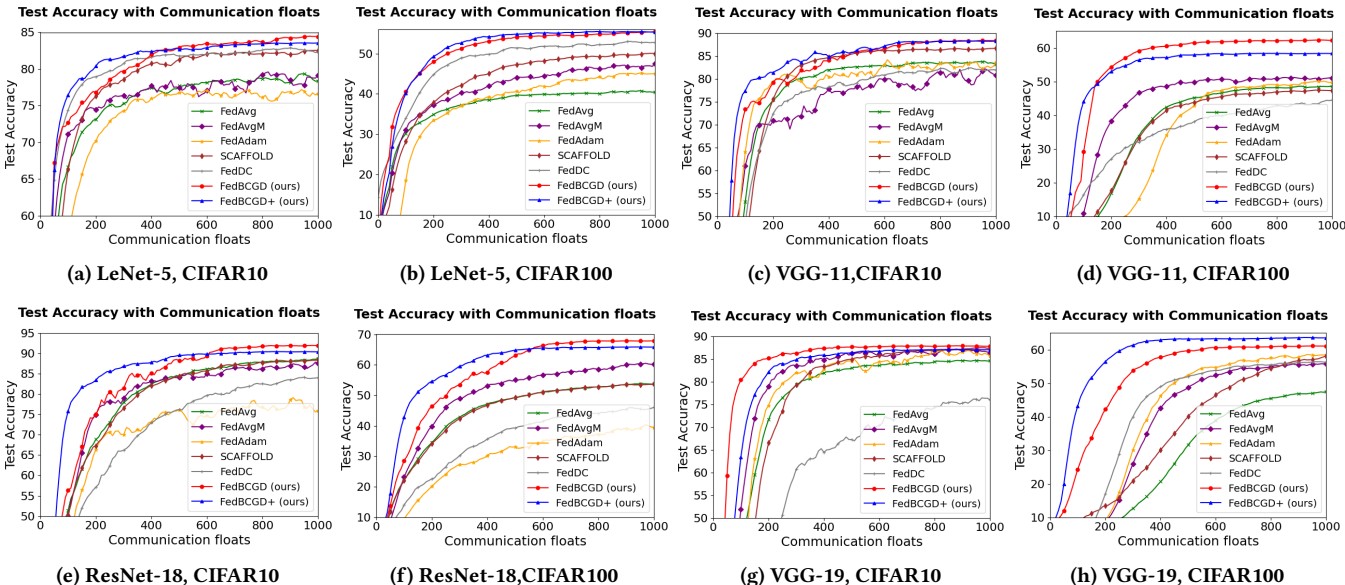

(a) LeNet-5, CIFAR10  (b) LeNet-5, CIFAR100  (c) VGG-11, CIFAR10  (d) VGG-11, CIFAR100

(e) ResNet-18, CIFAR10  (f) ResNet-18, CIFAR100  (g) VGG-19, CIFAR10  (h) VGG-19, CIFAR100

**Figure 3: The convergence comparison of our FedBCGD and FedBCGD+, and other baselines on the CIFAR10 and CIFAR100 datasets with different neural network architectures, where, in 100 clients, partial (10%) clients are used, and the heterogeneity parameter is set to $\rho = 0.6$.**

## 5.2  Results on Non-Convex Problems

**Results on Convolutional Neural Network:** From Tables 2 and 3, and Figure 3, we have the following observations: (i) Compared to FedAvg and its accelerated algorithms, FedBCGD significantly

reduces the communication floats per round, converges faster, and achieves more robust final model performance. In the experiment of LeNet-5 on CIFAR100, FedBCGD ($77d$) achieve 7.3× speedup to reach 40% accuracy, compared to FedAvg ($558d$). (ii) FedBCGD+

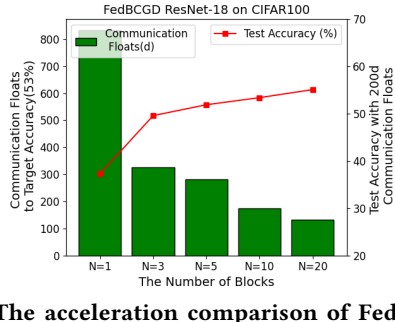

**Figure 4: The acceleration comparison of FedBCGD with different numbers of blocks.**

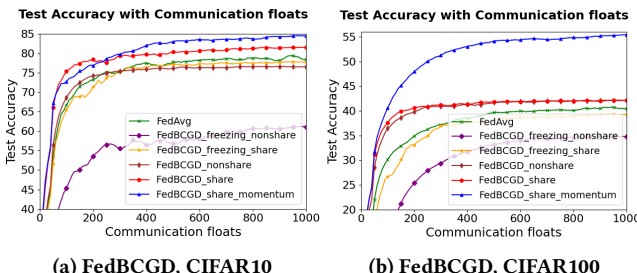

**(a) FedBCGD, CIFAR10**            **(b) FedBCGD, CIFAR100**

**Figure 5: Accuracy comparison of FedBCGD with LeNet-5 on CIFAR10 (a) and CIFAR100 (b), where heterogeneity is $\rho = 0.6$. FedBCGD_freezing_nonshare is updated by using the local freezing parameter algorithm without the shared block. FedBCGD_freezing_share refers to the FedBCGD_freezing algorithm with shared parameters. FedBCGD_nonshare is an algorithm that trains all parameters locally and only transmits parameter blocks during the upload process without shared parameters. FedBCGD_share refers to the algorithm that has shared parameters. FedBCGD_share_momentum (i.e., FedBCGD) refers to the algorithm that has momentum acceleration.**

further improves the convergence speed by client drift control and variance reduction, accelerating FedBCGD training process in experiments. In the experiment of ResNet-18 on CIFAR100, FedBCGD+ ($154d$) achieves $1.8\times$ speedup to reach 54% accuracy, compared to FedBCGD ($277d$). However, in terms of the final testing accuracy, it does not outperform FedBCGD. This means that FedBCGD+ has a faster convergence speed, requiring less communication floats at the specified accuracy, while the higher accuracy of our FedBCGD algorithm ultimately means that it has better generalization ability. And the generalization ability of our FedBCGD framework is better than those of other algorithms, e.g., FedAvg. (iii) The final accuracy of FedBCGD is much higher than that of Centralised SGD, which means that our FedBCGD has better generalization performance. That is, FedBCGD and FedBCGD+ can jump from a poor local minimum and converge to sharp local minima.

Figure 4 compares the effects of different block numbers under the same settings. When the number of blocks is 1, it degenerates into the FedAvgM algorithm. At the specified testing accuracy 53%, when the number of blocks is 20, our FedBCGD algorithm requires the least communication floats. The FedBCGD algorithm with 20

blocks achieves the highest accuracy with the same communication floats $200d$. As the number of blocks increases, the acceleration effect of the FedBCGD algorithm becomes more obvious.

From Figure 5, we can observe that freezing parameters in local training will cause client parameters to drift (purple line), resulting in poor performance. In addition, uploading parameters with shared parameters can improve convergence speed and final performance of the model (red line). Adding momentum compensation to client aggregation does accelerate convergence significantly (blue line).

**Table 4: Comparison of each algorithm on CIFAR100 and CIFAR10. Heterogeneity is $\rho = 0.1$, total communication floats are $1000d$, and the number of blocks in ResNet-18 is $N = 5$.**

| $\rho = 0.1$ | CIFAR100 (45%) | CIFAR10 (78%) |
|---|---|---|
| FedAvg [28] | $45.8 \pm 0.3$ ($741d$) | $78.1 \pm 0.4$ ($952d$) |
| FedAvgM [14] | $48.3 \pm 0.6$ ($769d$) | $78.6 \pm 0.8$ ($997d$) |
| FedAdam [32] | $49.9 \pm 0.5$ ($610d$) | $71.4 \pm 1.1$ ($\infty$) |
| SCAFFOLD [17] | $44.3 \pm 0.3$ ($\infty$) | $76.3 \pm 1.4$ ($\infty$) |
| FedDC [9] | $46.6 \pm 0.8$ ($278d$) | $79.1 \pm 0.8$ ($948d$) |
| **FedBCGD (ours)** | $\mathbf{59.5 \pm 0.3}$ (**147d**) | $\mathbf{86.2 \pm 0.9}$ (**212d**) |
| **FedBCGD+ (ours)** | $\mathbf{59.9 \pm 0.4}$ ($200d$) | $80.2 \pm 1.3$ ($768d$) |

**Table 5: The test accuracy comparison of each algorithm with ViT-Base on CIFAR100 and Tiny ImageNet. Heterogeneity is $\rho = 0.6$, total communication floats are $100d$, $N = 6$.**

| $\rho = 0.6$ | CIFAR100 (88%) | Tiny Imagenet (70%) |
|---|---|---|
| Centralised SGD | $81.5 \pm 0.3$ | $76.7 \pm 0.2$ |
| FedAvg [28] | $90.4 \pm 0.1$ ($24d$) | $71.2 \pm 0.1$ ($67d$) |
| FedAvgM [28] | $88.7 \pm 0.3$ ($32d$) | $76.7 \pm 0.4$ ($10d$) |
| FedAdam [32] | $87.6 \pm 0.2$ ($\infty$) | $65.5 \pm 0.6$ ($\infty$) |
| SCAFFOLD [17] | $88.2 \pm 0.3$ ($88d$) | $56.8 \pm 1.1$ ($\infty$) |
| FedDC [9] | $85.8 \pm 0.4$ ($25d$) | $55.0 \pm 1.2$ ($\infty$) |
| **FedBCGD (ours)** | $\mathbf{92.0 \pm 0.2}$ (**7d**) | $\mathbf{83.5 \pm 0.2}$ (**5.8d**) |
| **FedBCGD+ (ours)** | $90.6 \pm 0.3$ ($14d$) | $81.3 \pm 0.2$ (**4.6d**) |

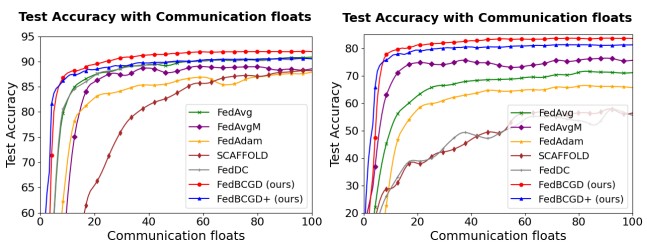

**(a) ViT-Base, CIFAR100**            **(b) ViT-Base, Tiny ImageNet**

**Figure 6: The test accuracy varies with the communication floats with ViT-Base on the CIFAR100 and Tiny ImageNet datasets, where $E = 1$ and $\rho = 0.6$ (best viewed in color).**

From results in Table 4, we compare the convergence speed of our algorithms and baseline algorithms under high levels of data heterogeneity. It can be observed that when data heterogeneity is high (e.g., $\rho = 0.1$), FedAvg converges slowly and struggles to reach the optimal point. In contrast, our algorithms consistently

converge and achieve better model generalization. Moreover, under high data heterogeneity, FedBCGD+ slightly outperforms FedBCGD, demonstrating the effectiveness of the variance control strategy in our FedBCGD+ algorithm.

In our experiments we get a phenomenon that our algorithm, FedBCGD and FedBCGD+, may generalizes better than Centralized SGD when the client data is not highly heterogeneous. The same phenomenon was also found in the literature [12, 23]. For highly non-convex problems, gradient decent and SGD methods are usually prone to fall into local minima, whereas the distributed methods local SGD are more prone to jump out of the local and sharp minimum and usually have better generalization ability [12].

**Results on Communication-efficient FL:** In Table 6, the FedBCGD algorithm outperforms the traditional efficient federated learning algorithms TOP-k and FedPAQ in terms of convergence speed and final generalization accuracy. The convergence can be further accelerated when the quantization strategy of QSGD is added to the chunks of FedBCGD.

Table 6: The test accuracy comparison of each algorithm with LeNet-5 on CIFAR100 and CIFAR10. Here, the heterogeneity is $\rho = 0.6$, total communication floats are $200d$, $N = 5$.

| $\rho = 0.6$ | CIFAR100 (40%) | CIFAR10 (70%) |
|---|---|---|
| FedAvg [28] | $35.4 \pm 0.1$ ($\infty$) | $73.2 \pm 0.1$ ($133d$) |
| TOP-k [1] | $42.2 \pm 0.5$ ($112d$) | $74.5 \pm 0.4$ ($92d$) |
| FedPAQ [33] | $43.3 \pm 0.2$ ($110d$) | $75.2 \pm 0.4$ ($121d$) |
| **FedBCGD (ours)** | $48.7 \pm 0.2$ ($91d$) | $\mathbf{77.2 \pm 0.2}$ ($65d$) |
| **FedBCGD+ (ours)** | $49.6 \pm 0.3$ ($89d$) | $80.6 \pm 0.2$ ($57d$) |
| **QSGD[2]+FedBCGD (ours)** | $52.2 \pm 0.4$ ($61d$) | $82.6 \pm 0.1$ ($32d$) |
| **QSGD[2]+FedBCGD+ (ours)** | $\mathbf{53.1 \pm 0.2}$ ($\mathbf{56d}$) | $\mathbf{83.2 \pm 0.3}$ ($\mathbf{29d}$) |

**Results on Vision Transformer:** To verify the effectiveness of our algorithm on large models, we adopt the most classic ViT-Base model on the Tiny ImageNet and CIFAR100 datasets. For the initialization of the model, we used the pretrained model downloaded from the official website. We divide the ViT-Base model into six parameter blocks. From the experimental results in Table 5 and Figure 6, we can observe that our FedBCGD algorithm can achieve the best results on the CIFAR100 dataset, and has more than 3× faster convergence speed, compared to FedAvg. The FedBCGD algorithm can achieve the best results on the Tiny ImageNet dataset, and attains more than 11.5× faster convergence speed. This can verify that FedBCGD can achieve excellent convergence speed on both Vision Transformer models and big datasets.

**Effectiveness of $\lambda$:** We tested FedBCGD using ResNet-18 on CIFAR100 dataset with momentum parameter $\lambda$ taking the values of {0.4, 0.5, 0.6, 0.7, 0.8, 0.9} and $\rho = 0.6$ . The convergence plots are shown in Figure 7. We note that setting $\lambda$ too small or too large impairs the convergence and generalization ability of FedBCGD. As shown in Figure 7, when $\lambda$ is relatively small, with $\lambda = 0.4$, the FedBCGD algorithm converges quickly, but the final generalization is not good. When we enlarge the value of $\lambda$, $\lambda = 0.8$, the convergence is slower but the final generalization is good. Empirically, we find the best performance is achieved when the $\lambda$ is set to around 0.8.

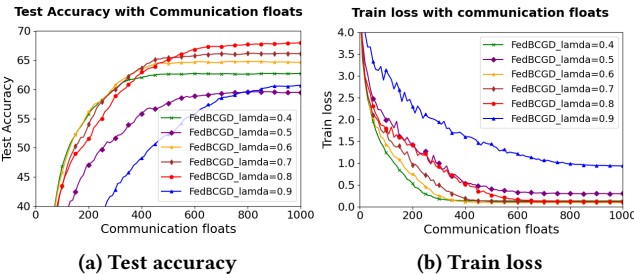

(a) Test accuracy      (b) Train loss

Figure 7: Test accuracy (a) and training loss (b) with ResNet-18 on CIFAR100, where $E = 5$ and $\rho = 0.6$. The number of parameter blocks is set to $N = 5$.

## 5.3 Results on Convex Problems

We conducted the classification tests on the EMNIST (byclass) dataset on classical logistic regression problems:

$$f(x) = \frac{1}{N} \sum_{i=1}^{N} \log\left(1 + \exp\left(-b_i a_i^\top x\right)\right) + \frac{\lambda}{2} \|x\|^2, \qquad (8)$$

where $a_i \in \mathbb{R}^d$ and $b_i \in \{-1, +1\}$ are the data samples, and $N$ is their total number. We set the regularization parameter $\lambda = 10^{-4} L$, where $L$ is the smoothness constant.

From Figure 8 (a,b), we observe that our FedBCGD and FedBCGD+ algorithms demonstrate faster convergence speed. Particularly, under the strong convexity, our FedBCGD+ algorithm exhibits even faster convergence compared to our FedBCGD, which aligns with our theoretical analysis.

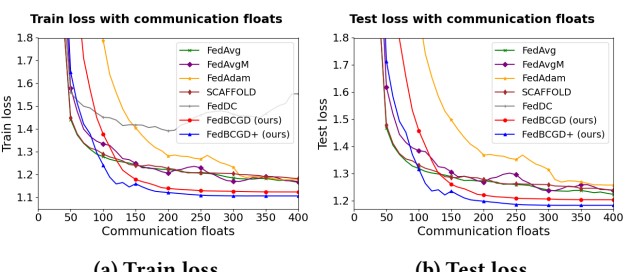

(a) Train loss      (b) Test loss

Figure 8: Logistic regression with $E = 1$ and $\rho = 0.1$. The number of blocks is set to $N = 5$.

## 6 CONCLUSION

This paper proposed the first federated block coordinate gradient descent method for horizontal federated learning. Moreover, we presented an accelerated version by using variance reduction and client parameter block drift control. In particular, we analyzed the convergence properties of the proposed algorithms, which show that our algorithms have significantly lower communication complexities than existing methods, and they also attain the best-known convergence rates for both convex and non-convex problems. Various experimental results verified our theoretical results and effectiveness of all the proposed algorithms. In the future, it is worthwhile to pay attention to how to more rationally divide model into blocks and how to choose the optimal parameter block to upload for clients.

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
