# OpenReview forum: "FedBCGD: Communication-Efficient Accelerated Block Coordinate Gradient Descent for Federated Learning"
_acmmm.org/ACMMM/2024/Conference — MM2024 Poster_

### Official Review · Reviewer_ktHw · 2024-05-21

**Rating:** 5
**Confidence:** 3

**Summary:**

This paper proposes a novel Federated Block Coordinate Gradient Descent (FedBCGD) method for communication efficiency. Specifically, FedBCGD splits model parameters into several blocks including a shared block and enables uploading a specific parameter block by each client during training. The analysis of convergence and extensive experiments validate the value of the proposed methods.

**Strengths:**

1. The paper is well written and easy to follow.
2. The approach is thoughtfully designed with a solid theoretical analysis.
3. Evaluations and comparisons with other methods validate the value of FedBCGD.

**Limitations:**

1. More comparisons with other Communication-efficient Federated Learning methods [1, 2, 3] should be conducted.
2. More datasets, especially in different modalities, should be used.

[1] Fedspeed: Larger local interval, less communication round, and higher generalization accuracy\
[2] Communication-Efficient Federated Learning with Accelerated Client Gradient\
[3] Communicationefficient adaptive federated learning.

**Suitability:**

2

---

### Official Review · Reviewer_XZ3H · 2024-05-22

**Rating:** 4
**Confidence:** 3

**Summary:**

This paper primarily discusses two federated learning frameworks, FedBCGD and FedBCGD+, which are used to improve communication efficiency. Given that federated learning typically operates under constrained communication resources, the paper is well-motivated. Specifically, the methods proposed involve dividing model parameters into multiple blocks and sharing some of these blocks locally to facilitate model aggregation on the server, thereby completing the federated learning process. Additionally, FedBCGD+ introduces a control variant to reduce local client drift, thus accelerating the convergence of federated learning. The benchmarks show that both FedBCGD and FedBCGD+ outperform the baselines presented in the paper.

**Strengths:**

I believe this is a very comprehensive paper that uses the proposed FedBCGD framework to address the issue of communication efficiency and improves the framework to further address the heterogeneity problem, resulting in FedBCGD+. The paper provides a thorough theoretical analysis of FedBCGD and demonstrates well-presented experimental results. In its theoretical analysis, the paper considers both strongly convex and non-convex scenarios, and in the experimental section, it validates the proposed method using logistic regression and DNNs, thereby proving its superiority over other baseline methods.

**Limitations:**

My primary concern is the statement of novelty:
1. The statement in the paper, "we employ stochastic gradient descent to update all parameters instead of parameter freezing during local training," is a method that has been well-studied and is frequently used in personalized federated learning, such as [1]. It would be advisable to clarify so as not to mislead readers into thinking this is a novelty of this paper.
2. The control variant has been introduced in SCAFFOLD to address data heterogeneity. To me, FedBCGD+ seems more like an integration of control variance variables into FedBCGD.
3. Additionally, the differences between BCD or FedBCD [2] and FedBCGD are not clearly explained in lines 168-171. I recommend including FedBCD in the benchmark in Table 1, as the concepts of FedBCD and FedBCGD are similar concerning parameter block updates.

[1] Pillutla, Krishna, et al. "Federated learning with partial model personalization." International Conference on Machine Learning. PMLR, 2022.
[2] Y. Liu et al., "FedBCD: A Communication-Efficient Collaborative Learning Framework for Distributed Features," in IEEE Transactions on Signal Processing, 2022,

Some minor issues:
1. Lines 208-213: It appears that one sentence is repeated.
2. The abbreviation 'FL' is introduced but the full term is still used in the text.
3. All citations in the reference list in the supplementary material are marked with question marks, suggesting issues with citation formatting.

**Suitability:**

3

---

### Official Review · Reviewer_dTwu · 2024-05-24

**Rating:** 4
**Confidence:** 3

**Summary:**

This paper presents a novel approach FedBCGD to improving communication efficiency in federated learning (FL) scenarios, particularly for large-scale models. The method reduces communication overhead by using block coordinate gradient descent, which is a novel contribution to the field of FL. Briefly, FedBCGD divides the model parameter into N blocks and a shared parameters block, each client only upload one of the N blocks and the shared block after local training. Differring from the BGD, the FedBCGD doesn’t freeze other parameters when update its own parameter block in a client. The authors also propose an accelerated version of the algorithm, FedBCGD+, which includes client drift control and stochastic variance reduction, indicating a deep exploration of the problem space. The paper also provides theoretical analysis of the proposed algorithms, including convergence properties, which adds to the quality of the research.

The problem of communication overhead in FL is significant, especially with the increasing size of models and data. The paper addresses this issue with potentially impactful solutions. The methods could be beneficial for applications in mobile intelligence, healthcare, and finance, where communication efficiency and data privacy are critical. The paper is structured logically and writes clearly. The proposed method appears to be the first of its kind to apply block coordinate descent to horizontal FL, which is an original contribution. And the accelerated version with variance reduction techniques, is also an innovative extension of the base algorithm.

**Strengths:**

1. Introduces a novel method for improving communication efficiency in FL.
2. Provides an accelerated version with additional improvements.
3. Offers a comprehensive theoretical analysis.
4. Experimental results demonstrate the effectiveness of the proposed methods.

**Limitations:**

1. The models in experiment is limited on classification tasks, the paper could benefit from a broader range of tasks to further validate the practical significance of the proposed methods.
2. The potential limitations of the proposed methods are not extensively discussed, which would be important for a balanced view.
3. The paper does not elaborate on how the block division strategy (not only the division number N) might affect different models or datasets, which could be an important consideration.

**Suitability:**

2

---

### Official Review · Reviewer_2ekS · 2024-05-25

**Rating:** 3
**Confidence:** 3

**Summary:**

This paper investigates block gradient descent for federated learning. The proposed FedBCGD method splits the model into multiple blocks and only updates a specific block to reduce the communication cost. Convergence analysis is also provided. Experiments are provided to demonstrate the outperformance.

**Strengths:**

The block-wise model update method is interesting.

Extensive theoretical analysis are provided for convergence analysis.

Diverse experiment evaluations and other comparisons are provided.

**Limitations:**

1 How to ensure the scalability and generalization of the proposed method? Is there any principle on model partition?

2 The model options are not convincing. The authors claimed “This paper is the first parameter block
communication work for training large-scale deep models”, while the largest model evaluated is the ViT-Base. At least a larger model should be considered.

3 The baselines are not sufficient, which are not sota. A few more sota methods should be evaluated, such as MOON and FedDyn for federated optimization, and PruneFL for communication efficiency.

4 Minor: Do the first/last blocks include an input layer/classifier?

**Suitability:**

2

---

### Meta-Review · Area_Chair_e6t7 · 2024-06-29

**Recommendation:** Accept (Poster)
**Confidence:** 5

**Metareview:**

The paper proposes a block coordinate gradient descent method to improve communication efficiency in federated learning. Despite some concerns about model scalability and generalization, the reviewers' feedback has been largely addressed in the rebuttal. The final score of the paper reads as 3x borderline accept and 1x weak accept. After careful consideration of the reviewers' assessments and responses, ACs concur with the reviewers and recommend accepting this submission. Congratulations. Please carefully consider the reviewers' feedback and suggestions for revision and incorporate promises made in the rebuttal.